# Socio-Economic Inequalities in Lung Cancer Outcomes: An Overview of Systematic Reviews

**DOI:** 10.3390/cancers14020398

**Published:** 2022-01-13

**Authors:** Daniel Redondo-Sánchez, Dafina Petrova, Miguel Rodríguez-Barranco, Pablo Fernández-Navarro, José Juan Jiménez-Moleón, Maria-José Sánchez

**Affiliations:** 1Instituto de Investigación Biosanitaria ibs.GRANADA, 18012 Granada, Spain; daniel.redondo.easp@juntadeandalucia.es (D.R.-S.); miguel.rodriguez.barranco.easp@juntadeandalucia.es (M.R.-B.); jjmoleon@ugr.es (J.J.J.-M.); mariajose.sanchez.easp@juntadeandalucia.es (M.-J.S.); 2Escuela Andaluza de Salud Pública, 18080 Granada, Spain; 3CIBER of Epidemiology and Public Health (CIBERESP), 28029 Madrid, Spain; pfernandezn@isciii.es; 4Cancer and Environmental Epidemiology Unit, National Center for Epidemiology, Carlos III Institute of Health, 28029 Madrid, Spain; 5Department of Preventive Medicine and Public Health, University of Granada, 18071 Granada, Spain

**Keywords:** lung cancer, socio-economic, disparities, inequalities, income, education, survival, treatment, diagnosis

## Abstract

**Simple Summary:**

Lung cancer is the leading cause of cancer mortality worldwide. Research has revealed significant socio-economic inequalities in lung cancer, such that people of lower socio-economic status (SES) generally have worse outcomes. In this article, using the umbrella review methodology, we review and organize the available evidence on socio-economic inequalities in diverse lung cancer outcomes. We find that people of a lower SES have a lower chance of cancer survival, most likely due to the lower likelihood of receiving both traditional and next-generation treatments, higher rates of comorbidities, and higher likelihood of being admitted as emergency. People of a lower SES are generally not diagnosed at later stages, but this may change after broader implementation of lung cancer screening, as early evidence suggests that there are socio-economic inequalities in its use.

**Abstract:**

In the past decade, evidence has accumulated about socio-economic inequalities in very diverse lung cancer outcomes. To better understand the global effects of socio-economic factors in lung cancer, we conducted an overview of systematic reviews. Four databases were searched for systematic reviews reporting on the relationship between measures of socio-economic status (SES) (individual or area-based) and diverse lung cancer outcomes, including epidemiological indicators and diagnosis- and treatment-related variables. AMSTAR-2 was used to assess the quality of the selected systematic reviews. Eight systematic reviews based on 220 original studies and 8 different indicators were identified. Compared to people with a high SES, people with a lower SES appear to be more likely to develop and die from lung cancer. People with lower SES also have lower cancer survival, most likely due to the lower likelihood of receiving both traditional and next-generation treatments, higher rates of comorbidities, and the higher likelihood of being admitted as emergency. People with a lower SES are generally not diagnosed at later stages, but this may change after broader implementation of lung cancer screening, as early evidence suggests that there may be socio-economic inequalities in its use.

## 1. Introduction

In 2020, lung cancer was the leading cause of cancer death worldwide, responsible for 18.2 percent of all cancer deaths, and the second most commonly diagnosed cancer for both sexes combined [1]. In men, lung cancer was the most commonly diagnosed cancer and the leading cause of cancer death [1]. In women, it was the third most commonly diagnosed malignancy and the second cause of cancer mortality [1].

Despite significant improvements in diagnosis and treatment in recent decades, lung cancer still has one of the lowest survival rates, with particularly low survival for individuals in advanced stages. Average five-year survival rates vary significantly between countries, with estimates ranging from 10% to 20% [2,3]. For instance, in Europe the average five-year age-standardized relative survival for lung cancer patients diagnosed between 2000 and 2007 was 13% according to EUROCARE-5 statistics [2,3].

Cancer outcomes are known to have an important social gradient and lung cancer is no exception. Factors such as socio-economic position, race, ethnicity, and place of residence have been found to produce social disparities in diverse cancer-related outcomes [4,5,6,7,8]. Among those, socio-economic status (SES) is a complex factor that can encompass several dimensions of the social and economic life circumstances of individuals and is often measured using information regarding education, income, and/or occupation [5].

Researchers have used diverse methodological approaches to the study of socio-economic disparities in cancer outcomes [5]. The individual approach uses information about each person’s socio-economic position, whereas the ecological approach uses area-based aggregated information such as deprivation indices that consider demographic, social, and/or economic factors of the residence area [5]. The magnitude and type of disparities identified could vary strongly from one approach to the other. However, both approaches offer useful and complementary information, and most importantly, both approaches have identified striking disparities in lung cancer outcomes including incidence [9], treatment [10,11], early mortality [12], and survival [11,13]. For instance, lung cancer incidence is higher among people from lower socio-economic status, as indicated by education, income, or occupation [14], using both individual and area-based indicators [9], with relative risks generally in the range of 1.5 to 1.8.

The last two decades have brought about important improvements in lung cancer diagnosis and treatment [15]. However, recent evidence suggests that people with a lower SES could be less likely to benefit from such improvements. For instance, in the US, lower SES is associated with lower rates of lung cancer screening [16,17] and lower rates of utilization of last-generation therapies [18,19].

Given the increasing number of outcomes for which socio-economic disparities have been investigated, the goal of the current research was to conduct an umbrella review of systematic reviews and map the evidence available regarding the role of SES in lung cancer outcomes, considering the methodological quality of the systematic reviews available and the quantity of the evidence.

## 2. Materials and Methods

We conducted an umbrella review (i.e., overview of systematic reviews) following methodological recommendations by several authors [20,21,22,23]. The unit of analysis in umbrella reviews is the systematic review and a preferred reporting items checklist for umbrella reviews is still under development [24]. Hence, in reporting the review, we followed the PRIO-harms checklist [25], a checklist designed for overviews of systematic reviews, including harms, by ignoring the items not applicable to the topic of our review (e.g., pertaining superficially to harms). A protocol was pre-registered in PROSPERO (ID: CRD42021282194), specifying the following criteria:

Population: Adult patients (18 years old or older) diagnosed with lung cancer.

Exposure/comparator: Socioeconomic status, specifically reviews comparing outcomes between patients with relatively lower vs. higher socioeconomic status. Any measure of socioeconomic status was considered, including individual-level and area-based measures (e.g., based on income, education, occupation, or a mixture thereof, area deprivation indices, etc.).

Outcomes: Diverse lung cancer outcomes were considered: epidemiological indicators (e.g., incidence, mortality, survival) and outcomes related to the diagnosis and treatment of lung cancer (e.g., screening, utilization of tests and treatments).

Study design: Systematic reviews of observational studies reporting on the relationship between socioeconomic status and any of the lung cancer outcomes considered.

Exclusion criteria: Articles published in 2009 or earlier (this criterion was set to exclude reviews that do not include the latest evidence); reviews that focus on other types of disparities (e.g., based on gender, race, rural vs. urban residence, etc.); reviews that do not report results specific to lung cancer; reviews that report on the prevalence of smoking or other risk factors or other outcomes not directly related to the diagnosis or treatment of lung cancer; reviews that are in a language not spoken by the research team (i.e., not English, Spanish, German, Dutch, Portuguese, Russian, Bulgarian), or the full text of which cannot be obtained.

Finally, we excluded reviews that were not considered systematic as per the definition of Martinic et al. [26]. As per this definition, the reviews were required to have the following characteristics to qualify as systematic: (i) research question, (ii) sources that were searched, with a reproducible search strategy (naming of databases, naming of search platforms/engines, search date and complete search strategy), (iii) inclusion and exclusion criteria, (iv) selection (screening) methods, (v) critical appraisal and report of the quality/risk of bias of the included studies, (vi) information about data analysis and synthesis that allows the reproducibility of the results.

Search strategy: We searched Medline via PubMed, Scopus, and Web of Science from 2010 onwards until 26 October 2021. Google Scholar was also searched manually to locate dissertations, pre-prints, and other unpublished materials. There were no restrictions by language in the search. Additional studies were identified by reviewing the reference lists of relevant reviews identified from the search. The full search strategy is available as Appendix A on The Open Science Framework (OSF) (http://doi.org/10.17605/OSF.IO/TS5G8 accessed on 13 January 2022).

Article selection and data extraction: The software Covidence (www.covidence.org, accessed on 13 January 2022) was used to manage all the stages of the review. Articles were screened based on title and abstract by two reviewers (D.P. and D.R.-S.). At this stage, articles receiving two “no” responses were excluded, whereas the rest (one or two “yes” responses) proceeded to full text review. Two reviewers (D.P. and D.R.-S.) read the full text of each article against the inclusion and exclusion criteria and disagreements were resolved through discussion. The reason for exclusion was recorded for each article.

Data was extracted from the articles selected for inclusion using a pre-defined form in Covidence by one reviewer (D.P.) and was then thoroughly checked by another reviewer (D.R.-S.). Per protocol, these data included: article title, first author, year of publication, type of measure of socioeconomic status used, lung cancer outcome studied, inclusion and exclusion criteria for original studies, dates covered by the literature search, number of original studies (k) and participants (n) included in the review, main results regarding the outcomes of interest (meta-analytic results with confidence interval estimates, *p*-values, and I^2^ values were extracted whenever available), and main limitations of the review. In addition to the data items pre-specified in the protocol, we extracted information about the geographical representation of the included studies, the reported funding of each review, whether the authors declared potential conflicts of interest, what instruments were used to evaluate the methodological quality of the original studies included in each review, and whether the reported analysis and conclusions were based on adjusted or unadjusted models.

Risk of bias: The risk of bias of the included reviews was assessed independently by two authors using the AMSTAR-2 checklist [27]. Disagreements were resolved through discussion. The AMSTAR-2 consists of 16 items that assess the rigor of systematic reviews of non-randomized studies on several domains, from which we considered five as critical for the current context: (i) whether a protocol was registered before the start of the review, (ii) adequacy of the literature search, (iii) risk of bias from individual studies included in the review, (iv) appropriateness of the meta-analytical methods used, and (v) consideration of this risk of bias in the interpretation of the results [27]. The results of individual items are not meant to be combined in a total score, instead, overall confidence in each review can be judged (e.g., high, moderate, low) based on weaknesses detected in critical and non-critical items [27]. We rated the reviews as follows: “high confidence” reviews had no weaknesses on any of the five critical domains and had no or only one weakness in a non-critical domain; “moderate confidence” reviews had no weaknesses on any of the critical domains but could have more than one non-critical weakness; and “low confidence” reviews had more than one critical weakness regardless of the number of non-critical weaknesses.

Data synthesis: Meta-analysis was not planned (nor meaningful for such a diverse set of outcomes). Results were synthetized narratively, considering the amount of the available evidence and the risk of bias of each review. In case of substantial overlap in the included original studies between two reviews (e.g., an older and more recent review on a very similar topic), the more recent and complete review was to be given more weight in the results.

## 3. Results

The flowchart of article selection is displayed in Figure 1. From 267 initial records, eight systematic reviews were included. The main reasons for exclusion for each article at the full-text stage are available as Appendix A on OSF (http://doi.org/10.17605/OSF.IO/TS5G8, accessed on 13 January 2022) (some articles could be excluded for more than one reason, but the software only permitted recording one).

Table 1 reports the main characteristics and results of the included studies and Table 2 reports information about the covariate adjustment, risk of bias assessment, and methodological limitations of each review. Appendix A on OSF (http://doi.org/10.17605/OSF.IO/TS5G8, accessed on 13 January 2022) reports additional review information including detailed inclusion and exclusion criteria, review funding sources, and conflicts of interest.

Overall, the eight systematic reviews reported on 220 non-overlapping original studies examining eight types of outcomes, including mortality, survival, screening utilization, cancer care interval duration, emergency presentation, stage at diagnosis, test use, and treatment receipt. Almost all original studies included in the reviews stemmed from high income countries (e.g., US, UK, European Union, and Australia).

The risk of bias ratings for each item of the AMSTAR-2 is displayed in Figure 2. Overall, four reviews were classified as “moderate confidence” and four as “low confidence”. 

### 3.1. Incidence and Mortality

We did not locate any reviews that met the inclusion criteria regarding incidence.

Wang et al. [28] investigated if childhood SES, defined as the education level or socioeconomic position of parents, and/or childhood housing conditions, influenced lung cancer mortality (number of studies k = 13). Lower childhood SES was associated with higher lung cancer mortality, HR = 1.25 (95% CI, 1.10, 1.42), *p* = 0.04, I^2^ = 49%, k = 10 (from adjusted analysis, for lowest vs. highest SES). There was also evidence for a dose–response relationship. This review was still in the pre-print stage and had not been externally reviewed. It received a low confidence rating in our assessment mainly because the pre-registered protocol did not match the reported review and the risk of bias of individual studies was not considered when interpreting the results (although most studies had good scores according to the scale used (see Table 2)).

### 3.2. Diagnosis-Related Outcomes

Screening: Sosa et al. [17] investigated socio-economic disparities in lung cancer screening, focusing only on studies conducted in the US (k = 6). Annual LCS with LDCT had been recommended in the US for adults aged 55 to 80 years who are current smokers with a 30 pack-year smoking history or who are former heavy smokers who have quit within the past 15 years [29] (these criteria have been recently updated to be broader [30]). In one available study, lower household income was associated with lower screening eligibility (k = 1). The effects of education on screening eligibility were mixed (k = 2). There was some evidence that patients with lower income were less likely to complete screening or have an intention to be screened (found in k = 2 out of k = 3). However, there was no difference in stage at diagnosis as a function of SES (k = 1). One study also found that high risk smokers with higher education had lower cancer-specific mortality. Overall, this review shows some evidence for socio-economic inequalities in screening in the US, although the number of included studies is very limited. This review received a low confidence rating mainly because no protocol was pre-registered and the risk of bias assessment was not strongly considered when interpreting the results (although the majority of studies had good scores on the assessment used, see Table 2).

Intervals in the cancer care pathway: Forrest et al. [31] found no differences in the duration of different intervals (eight intervals in total, such as patient, GP-referral, and diagnostic interval, among others) based on socio-economic status (k = 12). However, very few of the studies considered the patient’s health status, which is an important confounder because it can influence the time patients spend in the different healthcare intervals.

Emergency presentation: Forrest et al. [31] also investigated several proxy measures for delayed presentation such as acute presentation, emergency admission, number of times to consult, and diagnosis at death (k = 8). More deprived patients were more likely to present and/or be admitted as an emergency, but socio-economic inequalities were not found in the number of times to consult or in diagnosis at death.

Mitchel et al. [32] investigated what factors are related to emergency presentation defined as a diagnosis of cancer that arose during an unscheduled, emergency, or unplanned hospital admission. Higher socio-economic deprivation (as measured by area-based indices) was found to increase the likelihood of emergency presentation in all k = 3 studies identified. One study examined the association between annual household income and emergency presentation, but it was not significant.

As a result of the overlapping research questions, the studies identified by Mitchell et al. [32] and Forrest et al. [31] largely overlapped and reached similar conclusions. In particular, three of the four identified studies in the earlier review by Mitchell et al. were included in the later review by Forrest et al. Mitchell et al. received a low and Forrest et al. a moderate confidence rating. Overall, a yet small number of studies supports the idea that patients of lower SES are more likely to present and be admitted as an emergency.

Stage at diagnosis: Forrest et al. [31] found no evidence (k = 23) of socio-economic inequalities in late stage at diagnosis in the most, compared with the least, deprived groups (OR = 1.04, 95% CI = 0.92 to 1.19, *p* = 0.53, I^2^ = 60%, k = 7). Studies that were not suitable for meta-analysis (k = 16) showed the same pattern of results.

### 3.3. Treatment-Related Outcomes

Forrest et al. [10] investigated if SES was related to the receipt of traditional treatment (surgery, chemotherapy, radiotherapy, or unspecified treatment). Of the 46 studies they located, only one was based on an individual-level measure, whereas 45 used area-based measures of SES. Overall, lower SES was associated with lower likelihood of receiving any type of treatment: OR = 0.79 [95% CI 0.73 to 0.86], *p* < 0.001, I^2^ = 77%, k = 26. Regarding the different treatment modalities, patients of lower SES were less likely to undergo surgery: OR = 0.72 (95% CI 0.65 to 0.80), *p* < 0.001, I^2^ = 80%, k = 16 (also when controlling for histology and stage at diagnosis), chemotherapy, OR = 0.81 (95% CI 0.73 to 0.89), *p* < 0.001, I^2^ = 68%, k = 10, and unspecified treatment (i.e., when the article did not report the specific type of treatment), OR = 0.78 (95% CI 0.74 to 0.83), *p* < 0.001, I^2^ = 0, k = 5. There was no association of SES with the receipt of radiotherapy, OR = 0.99 (95% CI 0.86 to 1.14), *p* = 0.89, I^2^ = 54%, k = 7. One limitation of the review was that not all studies reported details about stage, histology, and comorbidity, all of which can influence treatment. This review received a moderate confidence rating and was the review with the fewest methodological flaws according to the AMSTAR-2.

Norris et al. [19] investigated if SES was related to the receipt of next generation treatments, including predictive biomarker tests and biological and precision therapies, locating k = 11 studies on lung cancer. There was no relationship between SES and utilization of predictive biomarker tests (epidermal growth factor receptor and/or anapestic lymphoma kinase): OR = 0.92 (95% CI 0.35 to 2.40), I^2^ = 97%, k = 2 (the two studies found relationships in the opposite direction). However, lower SES was related to lower utilization of biological and precision therapies, OR = 0.71 (95% CI 0.51 to 1.00), I^2^ = 95%, k = 6 (low compared to high SES). All studies identified were conducted in the US. An important limitation was that ORs were determined from raw data unadjusted for confounders. This review received a low confidence rating, mainly because of this methodological shortcoming and because the risk of bias of individual studies was not taken into account when analysing and interpreting the results.

Overall, results suggest that patients of lower SES are less likely to receive both conventional treatments such as surgery and chemotherapy and next-generation treatments such as biological and precision therapies.

### 3.4. Survival

Finke et al. [11] investigated socio-economic differences in lung cancer survival based on 94 studies (23 based on individual SES measures and 71 on area-based SES measures). Regarding individual-level measures, there was no effect of education level (HR = 1.03 [95% CI 0.96 to 1.10], ref = high education, I^2^ = 55%, k = 13) and occupations associated with lower SES did not show lower survival (k = 3). However, lower income was associated with lower survival (HR = 1.13 [95% CI 1.08 to 1.19], ref = high income, I^2^ = 0%, k = 7).

Regarding area-based measures, although meta-analysis could not be conducted the results were generally quite consistent, with lower survival among patients residing in lower SES areas, as shown by indicators of education (k = 3), income (k = 19), or a SES-index (k = 22).

Overall, studies were very heterogeneous, not only in the use of socioeconomic measures and aggregated levels but also in the reporting of survival measures and in the level of adjustment. Nevertheless, the range of differences between survival rates for lowest and highest SES groups was larger in studies considering area-based SES than in studies assessing individual SES. The generalizability of results to low-income countries is limited. This review received a moderate bias rating due to several methodological shortcomings that we did not consider critical.

Finally, Afshar et al. [6] conducted a large review to investigate the factors that explained socio-economic disparities in cancer survival, locating k = 7 studies that provided data on lung cancer. They concluded that inequalities in lung cancer survival appear to be partly explained by the receipt of treatment and comorbidities. Results regarding the mediating role of stage at diagnosis were mixed. A limitation was that the majority of studies used overall survival as outcome instead of cancer-specific survival (controlling for other causes of death), which would be more useful for studying potential mediators in the context of lung cancer, given that there are relevant socio-economic disparities regarding multiple causes of death. This review received a moderate bias rating due to several methodological shortcomings that we did not consider critical.

## 4. Discussion

There is evidence of multiple and multilevel socio-economic inequalities in diverse lung cancer outcomes related to diagnosis, treatment receipt, and survival from lung cancer. Figure 3 offers a general summary of the evidence available regarding inequalities in lung cancer outcomes, including both the results from the systematic reviews that met our strict inclusion criteria and the results from other reviews discussed below that offer valuable complementary information.

No systematic reviews about the relationship between SES and lung cancer incidence met the inclusion criteria. However, we are aware of two previous reviews [9,14] that found evidence for higher risk of lung cancer among people from lower SES, regardless of the type of SES indicator used (individual or area-based). One review was published before 2009 [14] and both lacked risk of bias assessment for the individual studies [9,14]. Both reviews were, however, based on a large number of studies and considered whether individual studies adjusted for smoking. The older review considered studies from all over the world by also conducting meta-analysis, whereas the more recent review was focused on Europe. Overall, despite these differences and considering the limitations, both reviews provided evidence that lung cancer risk is higher among people from lower SES. In addition, the effect of SES on lung cancer incidence persisted, albeit diminished, after adjusting for smoking. In particular, smoking was estimated to explain 40–70% of the increased risk among people with lower SES [9,14]. The prevalence of smoking tends to be higher among the socio-economically disadvantaged [33], however, the fact that socio-economic disparities in lung cancer risk persist after accounting for smoking habits suggests that there are other contributing factors to the effect of SES on lung cancer incidence.

Regarding cancer mortality, one review associated lower SES during childhood with higher cancer mortality during adulthood [28]. This review was still in a pre-print stage and received a “low confidence” rating due to multiple flaws. However, its conclusions are supported by another rapid review [34] that located 13 studies on lung cancer, in which lower childhood SES was related to higher mortality, with this effect being attenuated after adjusting for adult SES. Overall, the available results on epidemiological indicators suggest that both childhood and adulthood socio-economic circumstances are likely to contribute to adult lung cancer risk, at least partially through cumulative exposure to smoking.

When it comes to the diagnostic process, no differences were observed between the duration of different intervals on the cancer care pathway as a function of SES [31]. However, interval duration is traditionally very diversely reported and analysed, making results difficult to summarize in a review [35]. In addition, results may be misleading if the patient’s health state is not taken into account, as it can influence the duration of the intervals (e.g., patients who present with more advanced or acute symptoms may be referred, diagnosed, and treated faster) [31]. Hence, the relationship between SES and interval duration on the lung cancer care pathway should be examined again in the future, when more and better-quality evidence is available.

**Figure 3 cancers-14-00398-f003:**
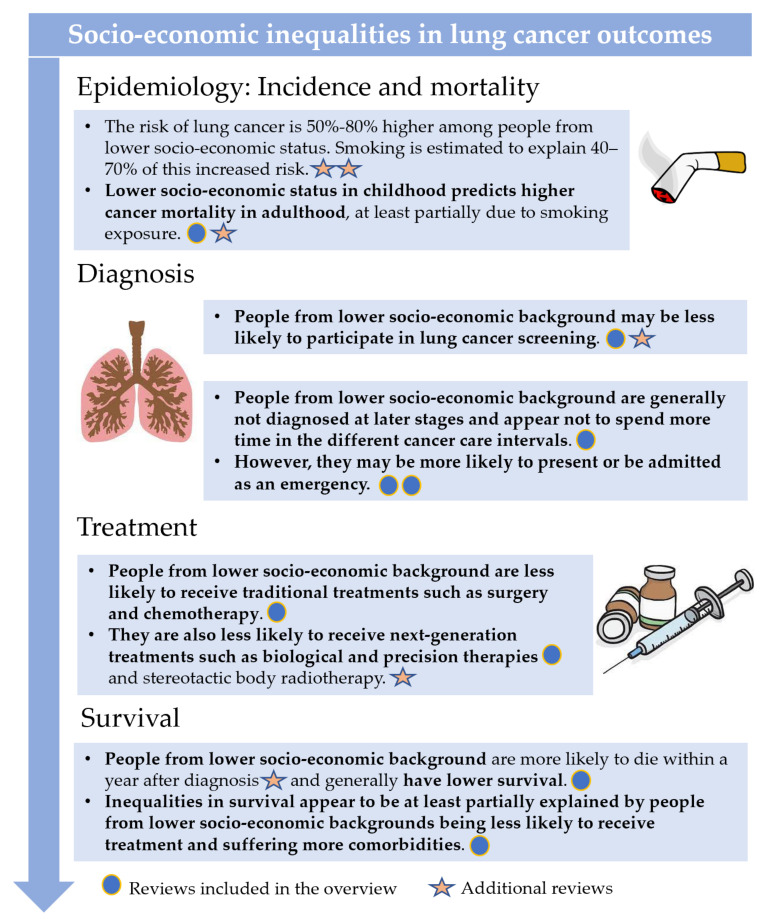
Summary of the available evidence about socio-economic inequality in lung cancer outcomes. Circles denote articles included in the overview of systematic reviews [6,10,11,17,19,28,31,32] and stars denote additional reviews [9,12,14,16,18,34].

In a similar vein, a somewhat more solid body of evidence documented no relationship between SES and stage at diagnosis [31]. The limited role of stage was also documented in other reviews, showing that associations between SES and surgery are independent of stage at diagnosis [10] and that, in contrast to some other cancers, the effect of SES on lung cancer survival is due to differences in treatment receipt and comorbidities, but not stage at diagnosis [6].

There was evidence that people from lower SES may be less likely to undergo screening for lung cancer [17] and may be more likely to be diagnosed as an emergency [31,32]. Both findings are based on a small number of studies, mostly from the US. For instance, lung cancer screening with low dose computed tomography (LDCT) is implemented in a very small number of countries (e.g., US, South Korea, Poland), but is already underway in many other countries (e.g., Canada, UK, Germany, Italy etc.) [36]. LDCT was found to reduce lung cancer mortality by 20–24% in high-risk smokers in the National Lung Screening Trial in the US [37] and the Dutch–Belgian NELSON trial in Europe [38]. However, with the current screening criteria, it may be that people of a lower SES are less likely to be eligible for screening [16] and appear to be underrepresented in screening programs [39]. Hence, we are yet to see the full extent of effects of socio-economic and other related factors (e.g., ethnicity, rurality, etc. [40]) on lung cancer screening outcomes. For instance, SES-based differences in stage at diagnosis could emerge in areas with population screening programs if there are underlying SES-based differences in screening eligibility and utilization, something that would widen the gap in survival between people with high vs. low SES.

A large body of evidence shows that individuals from lower socio-economic backgrounds have 20–30% lower odds of receiving traditional treatments, such as surgery and chemotherapy (with ORs between 0.70–0.80) [10]. A more recent review also concluded that they also had about 30% lower odds to receive next-generation treatments, such as biological and precision therapies [19]. However, this latter review was based on a much smaller number of studies conducted in the US only and on unadjusted effect estimates. Hence, more research is needed to confirm these findings and shed light on related issues, such as the use of predictive biomarker tests. For instance, in the treatment of non-small cell lung cancer, stereotactic body radiation therapy (SBRT) is a more recent treatment that is considered better than the more conventional fractionated external beam radiation (CFRT) (e.g., higher effectiveness and lower toxicity) [18]. A recent narrative review suggests that people from lower socio-economic backgrounds are less likely to receive SBRT and are instead more likely to receive CFRT o no treatment [18].

Overall, these results regarding the access to new screenings and therapies are generally in line with the inverse equity hypothesis [41], stating that new interventions, due to their elevated cost and other issues related to access, would initially only reach and benefit the rich, and thus widen socio-economic disparities in outcomes. For instance, more economically disadvantaged patients may be less likely to receive next-generation sequencing and thus be less likely to be eligible for new precision therapy clinical trials [42]. However, it is expected that as new interventions become standard practice, they would also reach the more socio-economically disadvantaged, eventually reducing socio-economic disparities [19,41].

Finally, perhaps the most abundant evidence in this context related lower SES to lower survival from lung cancer based on 94 studies from all over the world [11]. Studies were often too diverse to summarize into a common metric, but hazard ratios were generally not higher than HR = 1.10. Lower SES is also likely associated with early lung cancer mortality (broadly defined as death after 1–12 months following diagnosis [12]). In addition, another review specifically focused on exploring the underlying mechanisms by which SES affects cancer survival and concluded that, in the case of lung cancer, these were most likely explained by differences in treatment receipt and comorbidities [6].

The diverse socio-economic inequalities observed in multiple lung cancer outcomes suggest the need for a multi-level policy approach to reduce disparities that starts from primary lung cancer prevention and extends into survivorship. For instance, interventions that effectively increase awareness of lung cancer risk factors [43] and decrease tobacco use among socio-economically disadvantaged populations [33] can reduce inequalities in cancer incidence. Recently, several areas with “most immediate need for action” in the European context have been defined to reduce inequalities in cancer outcomes [44] that are highly relevant to lung cancer, in light of the results of the current review. These include equity, with regard to (1) awareness and compliance with the European Code against Cancer (i.e., a set of cancer prevention recommendations), (2) access to cancer screening programs, (3) reducing the time elapsed between the start of symptoms and the establishment of a cancer diagnosis, and (4) access to timely treatment and the treatment decision making process, among others.

Limitations of the current umbrella review include the perhaps very strict definition used to define a systematic review, which required that several criteria be met, among which that the search strategy is fully reported and that the individual studies are assessed in terms of risk of bias. On one hand, using such a strict definition resulted in the selection of the most robust and reproducible reviews. On the other hand, it meant that some otherwise excellent reviews which contain very useful findings were left out. To overcome this limitation, we have included such reviews in the overview of results displayed in Figure 3, which also clearly differentiates which reviews met the inclusion criteria and which did not.

None of the systematic reviews identified received a “high confidence” rating on the AMSTAR-2. We believe this is because the criteria used to assign such a rating were relatively strict and because many authors may not be aware of the importance of some methodological and reporting procedures in reviews of non-randomized observational studies. For instance, none of the reviews reported a list of the excluded studies with reasons for exclusion for each (item 7) and all but one failed to extract information regarding the funding of the original studies (item 10) and give justification for considering only non-randomized studies (item 3). Items 7 and 10 may be perceived as more relevant for reviews of clinical trials and item 3 as something self-explanatory because SES does not normally render itself to randomization. Overall, our opinion is that the reviews with a “moderate” rating have provided transparent and trustworthy synthesis of the data, whereas the findings of those with a “low” rating should be further confirmed, especially because these were also the reviews based on a relatively small number of studies. Hence, the overview is also limited by the quality and quantity of the evidence it is based on. As such, more evidence is needed in emerging areas of research such as screening and next-generation tests and treatments. Almost all of the evidence was based on high-income countries (e.g., USA, UK, other European countries). More studies are needed from middle- and lower-income countries to understand the socio-economic equalities that may exist within them.

Finally, only one of the identified reviews (Wang et al. on the effect of childhood SES on lung cancer mortality [28]) considered analysis stratified by sex, finding similar results for men and women. The rest of the reviews did not investigate to what extent the effect of SES varied among men and women, something that should be addressed in future research. The reviews on disparities in survival [6,11], conventional treatment receipt [10], and stage [31] were largely based on studies that controlled for sex in their primary analysis, so the presented results on these outcomes have been adjusted for this factor.

## 5. Conclusions

There are pervasive socio-economic inequalities in lung cancer incidence, mortality, and survival. These have been documented in both studies using individual-level indicators such as education, income, and/or occupation, and studies using area-level indicators such as neighbourhood deprivation indices. Compared to people with high SES, people with lower SES appear to be more likely to develop and die from lung cancer. People with lower SES also have lower cancer survival, most likely due to lower likelihood of receiving traditional and next-generation treatment, higher rates of comorbidities, and higher likelihood of being admitted as emergency. People with lower SES are generally not diagnosed at later stages, but this may change after broader implementation of lung cancer screening, as early evidence suggests that there are socio-economic inequalities in its use.

## Figures and Tables

**Figure 1 cancers-14-00398-f001:**
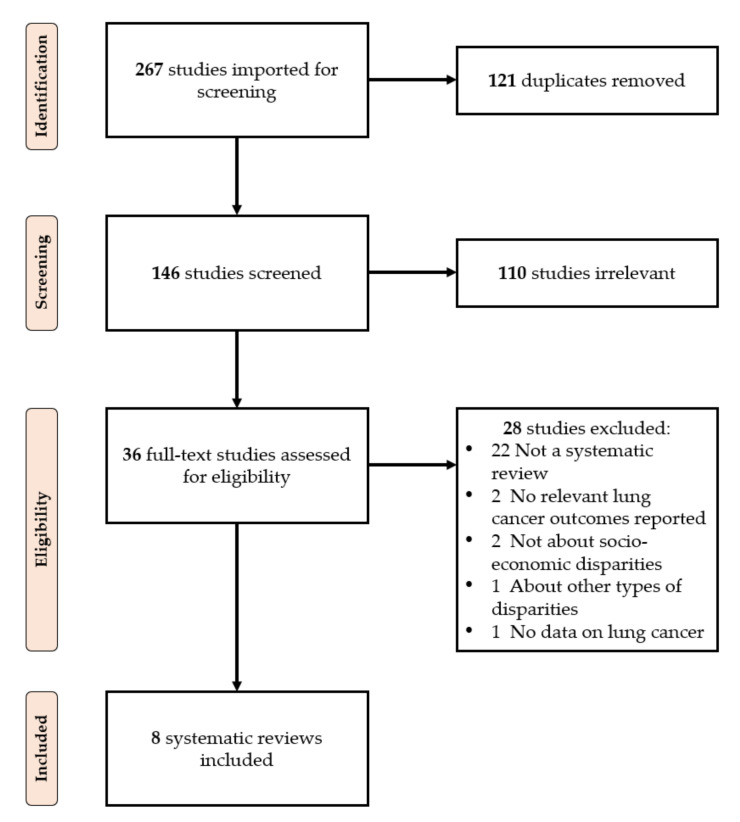
PRISMA flowchart of the review selection process.

**Figure 2 cancers-14-00398-f002:**
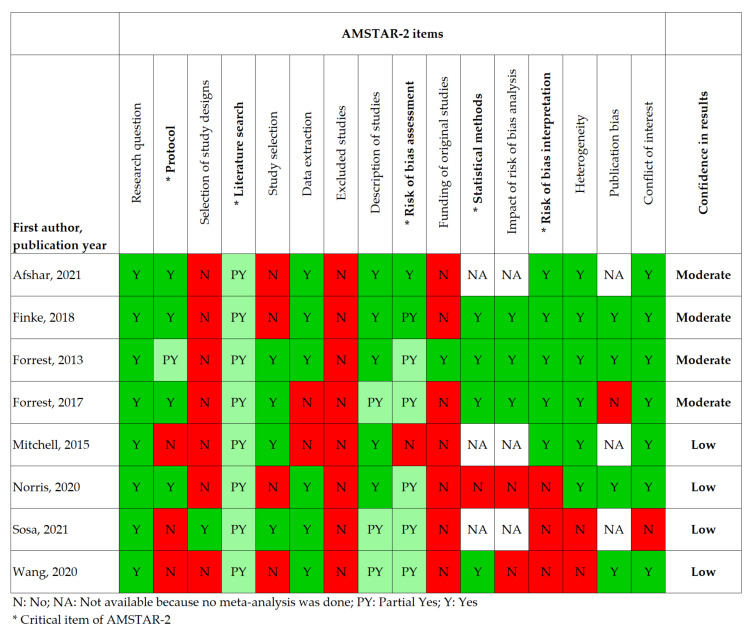
Risk of bias ratings for each item of the AMSTAR-2.

**Table 1 cancers-14-00398-t001:** Main characteristics and results of the included reviews.

First Author and Pub. Year	Measure(s) of SES	Lung Cancer Outcome(s)	Search Period	Number of Original Studies Included (k)	Geographical Representation of the Included Studies	Total Number of Participants/Patients	Main Results Regarding the Outcomes of Interest
Afshar 2021	Diverse individual and area-based	Survival (cancer-specific and overall)	2005–2020	7	UK (k = 3), Denmark (k = 1), Germany (k = 1), Sweden (k = 1), and US (k = 1)	111,275	Inequalities in survival appear to be partly explained by receipt of treatment and co-morbidities. Results regarding the mediating role of stage at diagnosis were mixed. No meta-analysis was performed.
Finke 2018	Diverse individual and area-based	Survival	Start–2017	94 (23 based on individual measures and 70 on area-based measures, 1 on both levels). 17 included in meta-analysis.	Individual measures: mostly from Scandinavia, US, and Italy; Area-based measures: mostly from US, UK, and Australia/New Zealand	>187,000 in studies with individual measures; >4,250,000 in studies with area-based measures.	Individual measures: There was no effect of education (HR = 1.03 (0.96, 1.10, ref = high education), I^2^ = 55%, k = 13). Lower income was associated with lower survival (HR = 1.13 (1.08, 1.19), ref = high income), I^2^ = 0%, k = 7). Occupations associated with lower SES did not show lower survival (k = 3), no meta-analysis. Area-based measures: Education (k = 3), income (k = 19), SES-index (k = 22). No meta-analyses could be conducted overall but results were generally consistent with lower survival among patients residing in lower SES areas. Comparison: The range of differences between survival rates for lowest and highest SES groups was larger in studies considering area-based SES than in studies assessing individual SES (Individual SES: range 1.0–12.8% units; area-based SES: range 0.9–22.9% units) but did not depend on the SES measure or the population size of the area.
Forrest 2013	Diverse individual and area-based	Receipt of treatment (surgery, chemotherapy, radiotherapy, unspecified treatment)	start–2012	46 (1 based on an individual measure and 45 on area-based measures of SES)	US (k = 18), UK (k = 19), Canada (k = 2), Sweden (k = 1), Australia (k = 1), Italy (k = 1), France (k = 1), and New Zealand (k = 3)	Surgery: >656,000 Chemotherapy: >359,000 Radiotherapy: >235,000 Unspecified treatment: >115,000	Overall: Lower SES was associated with lower likelihood of receiving treatment: OR = 0.79 [95% CI 0.73 to 0.86], *p* < 0.001, I^2^ = 77%, k = 26. Surgery: Patients of lower SES were less likely to undergo surgery: OR = 0.72 (95% CI 0.65 to 0.80), *p* < 0.001, I^2^ = 80%, k = 16 (also when controlling for histology and stage at diagnosis). Chemotherapy: Patients with lower SES were less likely to undergo chemotherapy, OR = 0.81 (95% CI 0.73 to 0.89), *p* < 0.001, I^2^ = 68%, k = 10. Radiotherapy: There was no association of SES with the receipt of radiotherapy, OR = 0.99 [95% CI 0.86 to 1.14], *p* = 0.89, I^2^ = 54%, k = 7. Unspecified treatment type: Patients with lower SES were less likely to receive treatment, OR = 0.78 [95% CI 0.74 to 0.83], *p* < 0.001, I^2^ = 0, k = 5.
Forrest 2017	Diverse individual and area-based	Stage (at diagnosis or start of treatment); duration of time intervals on the lung cancer pathway	start–2016	39 (23 for stage, 12 for intervals, and 8 for proxy measures of delay)	UK (k = 20), US (k = 10), Canada (k = 2), Denmark (k = 2), Sweden (k = 1), Australia (k = 1), Italy (k = 1), Korea (k = 1), and New Zealand (k = 1).	>267,500 for stage, >79,700 for intervals, >569,100 for proxy measures	Stage: There was no evidence of socio-economic inequalities in late stage at diagnosis in the most, compared with the least, deprived groups (OR = 1.04, 95% CI = 0.92 to 1.19, I^2^ = 60%, k = 7). Studies that were not suitable for meta-analysis (k = 16) showed the same pattern of results. Intervals: 12 studies examined 8 intervals from the cancer care pathway (e.g., patient interval, GP referral interval, diagnostic interval, etc.). There was no evidence of socio-economic inequalities. No meta-analysis was performed. Proxy measures of delay: These included acute presentation, emergency admission, number of times to consult and diagnosis at death (k = 8). More deprived patients were more likely to present and to be admitted as an emergency, but socio-economic inequalities were not found in the number of times to consult or in diagnosis at death.
Mitchell 2015	Diverse individual and area-based	Emergency presentation: a diagnosis of cancer that arose during an unscheduled (or emergency or unplanned) hospital admission.	1996–2014	4	UK (k = 3), US (k = 1)	163,050	Higher socio-economic deprivation (as measured by area-based indices) increased the likelihood of emergency presentation in all 3 studies identified. There was no association between annual household income and emergency presentation in the one study identified. No meta-analysis was conducted.
Norris 2020	Diverse individual and area-based (the large majority area-based)	Utilization of predictive biomarker tests and biological and precision therapies	1998–2019	11 (2 on predictive biomarkers and 9 on biological and precision therapies)	US (k = 11)	505,105 (15,588 for biomarker tests and 534,517 for therapy)	Predictive biomarkers: There was no relationship between SES and utilization of predictive biomarker tests (epidermal growth factor receptor and/or anapestic lymphoma kinase): OR = 0.92 (95% CI 0.35, 2.40), I^2^ = 97%, k = 2 (the two studies found relationships in the opposite direction). Biological and precision therapies: Lower SES was related to lower utilization of these therapies, OR = 0.71 (95% CI 0.51, 1.00), I^2^ = 95%, k = 6 (low compared to high SES).
Sosa 2021	Individual income and education	Screening eligibility; screening completion/intention; late stage diagnosis; lung cancer-specific mortality	2010–2020	6	US (k = 6)	163,418	Screening eligibility: There was evidence that higher household income was associated with greater screening eligibility (k = 1) but results regarding education were mixed (k = 2). Screening completion/intentions: There was some evidence that patients with lower income were less likely to complete screening or have the intention to be screened (found in k = 2 out of k = 3). Stage: There was no difference in stage at diagnosis as a function of SES (k = 1). Mortality: High risk smokers with higher education had lower cancer-specific mortality (k = 1).
Wang 2020	Childhood SES based on education level, socio-economic position of parents, and/or childhood housing conditions	Lung cancer-specific mortality	Start–2020	13 (8 in meta-analysis, 7 in dose–response analysis)	UK (k = 9), Norway (k = 3), Netherlands (k = 1)	2,779,242	Lower childhood SES was associated with higher lung cancer mortality, HR = 1.25 (95% CI, 1.10, 1.43), I^2^ = 49%, *p* = 0.04, k = unclear (from adjusted analysis). There was also a dose–response relationship.

**Table 2 cancers-14-00398-t002:** Information about covariate adjustment, risk of bias assessment, and methodological limitations of each review.

First Author and Pub. Year	Are Results and Conclusions Based on Adjusted Analysis in Original Studies?		Instrument Used to Assess the Methodological Quality of Original Studies	Overall Methodological Quality of Included Studies	Main Limitations of the Review
Afshar 2021	Yes	Studies adjusted for a variable combination of some of the following variables: Sex, age, stage, smoking, histopathology, treatment, comorbidity, performance status.	ROBINS-E (Risk of Bias in Non-Randomized Studies of Exposures) tool assessing confounding, selection of participants into the study, classification of the exposure, adjustment for mediators, level of missing data, measurement of the outcome, and reporting of results.	68 of 74 articles had critical risk of bias, the majority due to inappropriate adjustment.	Large between-study heterogeneity in the measures of SES and the methods used to identify the underlying causes of socio-economic inequalities. Frequent use of overall survival as outcome (instead of cancer-specific, which would be more useful for studying potential mediators in the context of lung cancer and considering competing risk of death).
Finke 2018	Yes	The majority of studies adjusted for age, gender, and stage. Some but not all studies also for smoking and treatment. Overall, adjustment was variable. Whenever possible, subgroup analyses were performed by adjustment for smoking status, stage, and treatment.	A modified version of the Newcastle-Ottawa-Scale (NOS) that assesses the quality of a study regarding the selection and comparability of study groups and ascertainment of the outcome (cohort studies) or exposure (case–control studies)	Mean quality scores of both individual and area-based studies were overall high, between 7 and 8 out of 8 possible points. The majority of studies used data from cancer registries, and hence many studies scored high on selection and outcome, adequacy of follow-up or representativeness of study population.	Studies were very heterogeneous, not only in the use of socioeconomic measures and aggregated levels but also in reporting of survival measures and in the level of adjustment. The generalizability of results to low-income countries is limited. No meta-analysis stratified by gender or subtype of lung cancer was possible.
Forrest 2013	Yes	The majority of studies adjusted for age, sex, and histology. However, it was variable and the ORs used in the meta-analyses were not consistently adjusted for the same covariates.	A study quality tool, adapted from existing quality tools (SIGN 50 and STROBE) that assessed population representativeness, internal validity, external validity, study reporting, confounding, and analysis (uni- or multi-variable).	Quality was variable but only studies conducting multivariable analysis (with high quality scores) were included in the meta-analysis.	Only one study measured SES on the individual level. Not all studies reported details of stage and histology—both of which influence treatment type—and some studies did not take comorbidity into account. The possibility for publication bias could not be excluded.
Forrest 2017	Partially	Stage: Only studies conducting multivariable analysis were included in the meta-analysis. However, the ORs used in the meta-analyses were not consistently adjusted for the same covariates. Other outcomes without meta-analysis: the presence of adjustment was variable.	A quality checklist adapted from Forrest et al., 2013 and the Aarhus checklist assessing validity and reliability of the outcome and exposure measures, analysis (adjustment and reporting), and study population.	No information on overall quality is reported. Only studies with sufficient quality were considered for meta-analysis (based on adjusted analysis).	Very few of the studies included in this current review took account of patient health status when examining time intervals. The included studies reported observational data only and use very diverse measures of socio-economic status. The meta-analysis may be underpowered to detect differences between early and late-stage presentation. The ORs used in the meta-analyses were not consistently adjusted for the same covariates. Many of the studies included in the narrative review were not of high quality. Publication bias is a possibility.
Mitchell 2015	Unclear	Nothing is reported in this respect.	Because included studies used methodologies that were often inapplicable to existing instruments, a method for assessing the strength of evidence of observational studies developed in previous systematic reviews was adapted, evaluating population, ascertainment, and analysis.	All studies reporting on SES were classified as “strong” (highest methodological classification).	Few evaluative studies were identified, with most researchers undertaking observational work utilizing routine data.
Norris 2020	No	Unadjusted ORs were used to enable inclusion of as many studies as possible in a consistent way (because not many original studies reported adjusted ORs)	A modified version of the ISPOR checklist for retrospective database studies assessing data sources, statistical results of interest, and generalizability of conclusions drawn.	Quality scores for the lung cancer studies were variable: ranging from 6 to 10 points for the biomarker studies and 6 to 8.5 for the biological and precision therapies (from a maximum of 10).	ORs were determined from raw data unadjusted for confounders. There was large heterogeneity between studies. Title and abstract screening were carried out by a single reviewer. There were only a few studies available, and most were from the US and in older patients.
Sosa 2021	Partially	All but one studies were judged as “adjusting statistically for confounders”; however, it is not specified with confounders.	The National Heart, Lung, and Blood Institute Quality Assessment Tool for Observational Cohort and Cross-Sectional Studies that assesses potential flaws in study methods or implementation that may lead to bias and lower study quality.	Five studies were judged as “good” (best category) and one as “fair” (second best).	Only a small number of studies was available; no gender-related disparities were reviewed; most studies were observational and/or secondary analysis studies; only a qualitative synthesis could be conducted.
Wang 2020	Yes	It is not reported what factors were adjusted for.	A score system based on the Newcastle-Ottawa Scale (NOS) tool assessing representativeness of exposure arm(s), selection of the comparative arm(s), origin of exposure source, demonstration that outcome of interest was not present at start of study, studies controlling the most important factors, studies controlling the other main factors, assessment of outcome with independency, adequacy of follow-up length (to assess outcome), lost to follow-up acceptable (less than 10% and reported).	Quality scores for the studies included in the meta-analysis ranged from 6 to 8 (out of maximum of 9), with 8 being the most frequent value.	The included studies do not cover diverse geographical regions and do not report data from recent years. Analyses based on cancer subtypes could not be conducted.

## Data Availability

All supporting data for the review can be downloaded from the Open Science Framework (OSF): http://doi.org/10.17605/OSF.IO/TS5G8, accessed on 13 January 2022.

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
