# Peer review of "Socio-Economic Inequalities in Lung Cancer Outcomes: An Overview of Systematic Reviews"

_cancers, 2022, doi:10.3390/cancers14020398_

Round 1
Reviewer 1 Report
The present study evaluates the impact of socio-economic status on lung cancer outcomes. The authors have performed a thorough selection of existing systematic-reviews on the topic using strict criteria for inclusion and analyzed the influence of socio-economic status on several parameters related to lung cancer incidence, diagnosis, treatment and survival. The text is easy to follow, clear and well written, with only minor English spell check required. The methodology is explained in detail and appropriate resources are listed. In my opinion, the manuscript is of interest to readers as it summarizes the findings regarding the influence of socio-economic status on lung cancer care, highlighting both significant results and potential limitations of included studies. Considering the high mortality rate of lung cancer patients, the increasing incidence worldwide and the high costs generated by these patients in the healthcare system, this study could serve as a motivation for a more aggressive screening of patients with lower socio-economic status in the hope of an earlier diagnosis and an improved outcome.
Author Response
Reviewer 1
The present study evaluates the impact of socio-economic status on lung cancer outcomes. The authors have performed a thorough selection of existing systematic-reviews on the topic using strict criteria for inclusion and analyzed the influence of socio-economic status on several parameters related to lung cancer incidence, diagnosis, treatment and survival. The text is easy to follow, clear and well written, with only minor English spell check required. The methodology is explained in detail and appropriate resources are listed. In my opinion, the manuscript is of interest to readers as it summarizes the findings regarding the influence of socio-economic status on lung cancer care, highlighting both significant results and potential limitations of included studies. Considering the high mortality rate of lung cancer patients, the increasing incidence worldwide and the high costs generated by these patients in the healthcare system, this study could serve as a motivation for a more aggressive screening of patients with lower socio-economic status in the hope of an earlier diagnosis and an improved outcome.
A: Thank you for the positive evaluation of our manuscript.

Reviewer 2 Report
Hi,
The authors have presented a Review examining the influence of social-economic standing in the incidence, mortality and survival among lung cancer patients. Overall, the authors have done a good job including the data since 2009 and factoring in the risk of bias. There are a few suggestions for the authors to address in their study.
- Is there a difference in the disease prevalence between the different economic strata, and if yes how does it change after smoking is adjusted?
- It is known that people economic restrictions have limited resources available to them. This is reflected in the disease outcome and presented in the manuscript. However, what measures can be taken to bridge this gap, or provide increased help to the people in the lower economic strata?
- The disparity caused by socio-economic differences holds true between both the sexes?
- How has advances in medicine, like new treatments and biosimilars, impacted this socio-economic induced bias?
Author Response
Reviewer 2
The authors have presented a Review examining the influence of social-economic standing in the incidence, mortality and survival among lung cancer patients. Overall, the authors have done a good job including the data since 2009 and factoring in the risk of bias. There are a few suggestions for the authors to address in their study.
A: Thank you for the positive evaluation of our manuscript.
- Is there a difference in the disease prevalence between the different economic strata, and if yes how does it change after smoking is adjusted?
A: We could not find reliable information regarding the prevalence of lung cancer as a function of SES. However, we think that information regarding the incidence (risk of developing lung cancer) can be useful to address this question. In particular, on p. 15 we have included the following information:
“Overall, despite these differences and considering the limitations, both reviews provided evidence that lung cancer risk is higher among people from lower SES. In addition, the effect of SES on lung cancer incidence persisted, albeit diminished, after adjusting for smoking. In particular, smoking was estimated to explain 40–70% of the increased risk among people with lower SES [9,14]. The prevalence of smoking tends to be higher among the socio-economically disadvantaged [33], however, the fact that socio-economic disparities in lung cancer risk persist after accounting for smoking habits suggests that there are other contributing factors to the effect of SES on lung cancer incidence.”
- It is known that people economic restrictions have limited resources available to them. This is reflected in the disease outcome and presented in the manuscript. However, what measures can be taken to bridge this gap, or provide increased help to the people in the lower economic strata?
A: We have now discussed some suggestions on page 17:
“The diverse socio-economic inequalities observed in multiple lung cancer outcomes suggest the need for a multi-level policy approach to reduce disparities that starts from primary lung cancer prevention and extends into survivorship. For instance, interventions that effectively increase awareness of lung cancer risk factors [43] and decrease tobacco use among socio-economically disadvantaged populations [33] can reduce inequalities in cancer incidence. Recently, several areas with “most immediate need for action” in the European context have been defined to reduce inequalities in cancer outcomes [44] that are highly relevant to lung cancer in light of the results of the current review. These include equity with regards to 1) awareness and compliance with the European Code against Cancer (i.e., a set of cancer prevention recommendations), 2) access to cancer screening programs, 3) reducing the time elapsed between the start of symptoms and the establishment of a cancer diagnosis, and 4) access to timely treatment and the treatment decision making process, among others.”
- The disparity caused by socio-economic differences holds true between both the sexes?
A: We have addressed this question in the limitations section on page 18:
“Finally, only one of the identified reviews (Wang et al. on the effect of childhood SES on lung cancer mortality [28]) considered analysis stratified by sex finding similar results for men and women. The rest of the reviews did not investigate to what extent the effect of SES varied among men and women, something that should be addressed in future research. The reviews on disparities in survival [6, 11], conventional treatment receipt [10], and stage [31] were largely based on studies that controlled for sex in their primary analysis, so the presented results on these outcomes have been adjusted for this factor.”
- How has advances in medicine, like new treatments and biosimilars, impacted this socio-economic induced bias?
A: We have discussed this on page 17:
“Overall, these results regarding the access to new screenings and therapies are generally in line with the inverse equity hypothesis [41], stating that new interventions, due to their elevated cost and other issues related to access, would initially only reach and benefit the rich, and thus widen socio-economic disparities in outcomes. For instance, more economically-disadvantaged patients may be less likely to receive next-generation sequencing and thus be less likely to be eligible for new precision therapy clinical trials [42]. However, it is expected that as new interventions become standard practice, they would also reach the more socio-economically disadvantaged eventually reducing socio-economic disparities [19, 41].”

Reviewer 3 Report
This is a very interesting approach to summarize the current knowlegde regaring socio-economic status as cancer care outcome determinant. The rarely used umbrella review used a very strict approachg according to the AMSTAR-2 criteria. The applicability of these critreria for reviews about outcome research and related factors is the most critical point in this review. Hoiwever, the authors were fully aware of this issue and discussed it appropriately. Overall, I congratulate to this work.
From my point of view one important aspect is still missing in the presentation of the included reviews. SES is intensively related to healthcare insurance topics, such as availibility, accessibility, affordibility and accaptability of care. This differs very much between countries. Therefore, it would be interesting to add information about the regions that were addressed in the reviews. The shortcomings and research challanges were adressed in the discussion, but should be also presented in the results.
An addisional minor point is figure 3. Here the authors referenced in almost half of the statements to reviews that were not included in the analysis before. This seems to be inconsistent.
Author Response
Reviewer 3
This is a very interesting approach to summarize the current knowlegde regaring socio-economic status as cancer care outcome determinant. The rarely used umbrella review used a very strict approachg according to the AMSTAR-2 criteria. The applicability of these critreria for reviews about outcome research and related factors is the most critical point in this review. Hoiwever, the authors were fully aware of this issue and discussed it appropriately. Overall, I congratulate to this work.
A: Thank you for the positive evaluation of our manuscript.
From my point of view one important aspect is still missing in the presentation of the included reviews. SES is intensively related to healthcare insurance topics, such as availibility, accessibility, affordibility and accaptability of care. This differs very much between countries. Therefore, it would be interesting to add information about the regions that were addressed in the reviews. The shortcomings and research challanges were adressed in the discussion, but should be also presented in the results.
A: We agree that this is a very important aspect and have included this information as suggested in a new column in Table 1 (Geographical representation of the included studies).
An addisional minor point is figure 3. Here the authors referenced in almost half of the statements to reviews that were not included in the analysis before. This seems to be inconsistent.
A: We realize that this may be unusual and have thus emphasized it in the manuscript, so that the differentiation, and hence the weight that should be assigned to the mentioned evidence, is clear to readers.
“Figure 3 offers a general summary of the evidence available regarding inequalities in lung cancer outcomes, including both the results from the systematic reviews that met our strict inclusion criteria and the results from other reviews discussed below that offer valuable complementary information.”
“On the other hand, it means that some otherwise excellent reviews which contain very useful findings have been left out. To overcome this limitation, we have included such reviews in the overview of results displayed in Figure 3, which also clearly differentiates which reviews met the inclusion criteria and which not.”
In addition, we have now highlighted the results stemming from reviews that met the inclusion criteria in bold, thus increasing the contrast between included and not included reviews in the figure. We believe that this will help readers clearly differentiate between the two.
